# A simple method for unsupervised anomaly detection: An application to Web time series data

**Keisuke Yoshihara**[1]*, **Kei Takahashi**[2,3]

**1** Center for Mathematics and Data Science, Gunma University, Maebashi, Gunma, Japan, **2** Faculty of Information Engineering, Fukuoka Institute of Technology, Fukuoka, Japan, **3** School of Statistical Thinking, The Institute of Statistical Mathematics, Tokyo, Japan

* ksk0110@gmail.com

## Abstract

We propose a simple anomaly detection method that is applicable to unlabeled time series data and is sufficiently tractable, even for non-technical entities, by using the density ratio estimation based on the state space model. Our detection rule is based on the ratio of log-likelihoods estimated by the dynamic linear model, i.e. the ratio of log-likelihood in our model to that in an over-dispersed model that we will call the NULL model. Using the Yahoo S5 data set and the Numenta Anomaly Benchmark data set, publicly available and commonly used benchmark data sets, we find that our method achieves better or comparable performance compared to the existing methods. The result implies that it is essential in time series anomaly detection to incorporate the specific information on time series data into the model. In addition, we apply the proposed method to unlabeled Web time series data, specifically, daily page view and average session duration data on an electronic commerce site that deals in insurance goods to show the applicability of our method to unlabeled real-world data. We find that the increase in page view caused by e-mail newsletter deliveries is less likely to contribute to completing an insurance contract. The result also suggests the importance of the simultaneous monitoring of more than one time series.

## Introduction

In recent years, more business entities have increased their advertising relevance because of a shift from conventional mass media advertising to digital advertising, which is associated with the emergence of new technology-driven advertising. In terms of advertising on Electronic Commerce (EC) sites, both physical (traditional) advertising agencies and web advertising agencies, the EC sites' owners, and production and management firms are involved. Under such circumstances, the information shared among relevant business entities is more likely to be asymmetric. For example, unexpected events are possible for web advertising agencies, such as a sudden increase in the number of visitors to an EC site, due to a TV commercial spot. Such events look like anomalies on unlabeled time series data from a web advertising agency's perspective. Detecting anomalies and pinning down their sources are particularly important

**Data Availability Statement:** The data sets used in the evaluation section are publicly available: the Yahoo S5 data set is provided as part of the Yahoo! Webscope program (https://webscope.sandbox.yahoo.com/catalog.php?datatype=s&did=70) and

the Numenta Anomaly Benchmark (NAB) data set is available from the following repository (https://github.com/numenta/NAB). The data set used in the application section cannot be shared publicly because of third party protection. The data set underlying the results presented in the application section is available from Flat inc.(https://www.flat-inc.jp/).

**Funding:** The authors received no specific funding for this work.

**Competing interests:** The authors have declared that no competing interests exist.

for less-informative entities. If the less-informative entities could detect anomalies in a timely fashion, they would be able to contact the relevant business entities regarding detected anomalies. For example, they can ask if some marketing campaigns are implemented. Reducing information disparity among relevant business entities is obviously crucial to achieving business success.

In the paper, we propose a simple anomaly detection method that utilizes the density ratio estimation and the state space model. We combine these two existing methods in consideration of (i) the applicability to unlabeled data, (ii) the applicability to time series data, and (ii) the ease of maintaining and repairing the model or system. The goal of this study is to detect anomalies quickly and precisely. For such objective, we employ the dynamic linear model (DLM), which enables us to achieve fast detection of anomalies and the systematic extension to multiple dimensions. The major difference from the existing methods is that we incorporate the time-series-specific information into the model, which enables us to detect anomalies precisely, while the existing methods basically utilize past observations only when predicting future observations. In addition, in time series anomaly detection, anomalous data at time $n$ usually influences the prediction at time $n + 1$. It leads to sharp fluctuation of the prediction and makes it difficult to detect anomaly at $n + 1$ accurately. Because our method uses Bayesian model, we can easily address such a problem. Furthermore, most of the important Key Performance Indicators (KPIs) in the marketing field, such as revenue, conversion rate and number of page views are often observed as unlabeled and non-stationary time series data. Our method is applicable to such data. Moreover, our method is based on the simple idea so that it is tractable enough for non-technical entities such as EC sites' owners. The anomaly detection rule is based on the ratio of log-likelihoods estimated by the DLM, i.e. the ratio of log-likelihood in our model to that in an over-dispersed model that we call the NULL model.

We evaluate the performance of our method using the Yahoo S5 data set and the Numenta Anomaly Benchmark (NAB) data set, publicly available benchmark data sets with labeled anomalies, and find that our method can achieve better or comparable performance compared to the existing methods. This is mainly because of the potential of our method to take into account larger number of explanatory variables in the model than the existing methods. Our evaluation results imply that it is essential in time series anomaly detection to incorporate the specific information on time series data into the model. Moreover, to show the applicability of our method to unlabeled real-world data, we apply it to unlabeled Web time series data, specifically, daily Page View (PV) and average Session Duration (SD) data on an EC site that deals in insurance goods for 2015–2018 from the perspective of a web advertising agency. By discussing the relationship between the detected anomalies and external factors to the Web advertising agency, e-mail newsletter deliveries, we find that the increase in PV caused by e-mail newsletter deliveries is less likely to contribute to completing an insurance contract. The result also suggests the importance of the simultaneous monitoring of more than one time series.

Anomaly detection has been widely applied in various fields such as intrusion and fraud detection [1]. However, to the best of our knowledge, there are few applications in marketing in spite of its growing importance in the digital marketing era. Bozbura et al. [2] present a comparative analysis of six commonly used anomaly detection methods in the context of e-commerce and point out the importance of anomaly detection in the marketing field. Anomaly detection is beneficial not only for reducing information disparities about marketing campaigns among relevant business entities, which is mentioned above, but also for monitoring and assessing the effectiveness of marketing activities, and preventing revenue loss due to anomalies that may arise from, for example, server failure and incorrect price entries.

Since the pioneering work of Fox [3], various methods on anomaly detection in time series data have been proposed in the literature, which include the method using an autoregressive

model [4], an ARIMA model [5, 6], a hierarchical temporal memory network [7], and Convolutional Neural Networks (CNNs) [8]. In particular, the literature on anomaly detection using a filtering method include Soule et al. [9], Manandhar et al. [10], and Nakano et al. [11]. The first two studies use the Kalman filter, but their methods are applied to labeled data only. The third one proposes a generalized exponential moving average model and introduces three anomaly detection methods. Because their model is estimated by a particle filter, it becomes intractable due to the curse of dimensionality when it is applied to multivariate time series data, while our method can systematically be extended to multivariate cases. There are two general streams in the literature on anomaly detection in multiple time series data: one applies a method to each time series independently, and the other deals with multiple time series simultaneously. In the former stream, for example, Hundman et al. [12] propose a method using the Long Short-Term Memory and apply it to each time series, but ignore the correlation between time series. Some other studies use dimensionality reduction techniques such as principal component analysis to obtain a new set of uncorrelated time series or univariate time series and apply a method to each new time series [13–17]. This study contributes to the latter stream, which includes the methods using, for example, the Contextual Hidden Markov Model [18] and the CNNs [8]. By virtue of the scalability of DLM, our method can also be applied to multivariate time series data, taking into account the correlation between time series without any additional pre-processing such as dimensionality reduction.

## Method

We consider the following DLM.

$$
\begin{aligned}
\boldsymbol{Y}_n &= \boldsymbol{F}_n \boldsymbol{X}_n + \epsilon_n, & \epsilon_n &\sim \mathcal{N}(\boldsymbol{0}, \boldsymbol{V}_n) \\
\boldsymbol{X}_n &= \boldsymbol{G}_n \boldsymbol{X}_{n-1} + \boldsymbol{\omega}_n, & \boldsymbol{\omega}_n &\sim \mathcal{N}(\boldsymbol{0}, \boldsymbol{W}_n)
\end{aligned}
$$

where $\boldsymbol{Y}_n \equiv \{Y_{1,n}, Y_{2,n}, \cdots, Y_{K,n}\}'$ is the observation at time $n$, $\boldsymbol{X}_n$ represents the state of the system at time $n$, $\epsilon_n$ and $\boldsymbol{\omega}_n$ are observation and system noises which follow the multi-dimensional Gaussian distribution with mean zero-vectors and covariance matrices $\boldsymbol{V}_n$ and $\boldsymbol{W}_n$, respectively, and $\boldsymbol{F}_n$ and $\boldsymbol{G}_n$ are known matrices. Note that our method does not restrict the resolution of the data. The variable $n$ can represent second, minute, hour, day, week, month, and so on. Suppose that we have $N$ observations $\boldsymbol{Y}_1, \cdots, \boldsymbol{Y}_N$ and its distribution depends on an unknown hyper-parameter $\boldsymbol{\theta}$. Then, as a function of $\boldsymbol{\theta}$, the likelihood function can be written as follows.

$$
L(\boldsymbol{\theta}) = p(\boldsymbol{Y}_1, \cdots, \boldsymbol{Y}_N; \boldsymbol{\theta}) = \prod_{n=1}^{N} p(\boldsymbol{Y}_n | \boldsymbol{Y}_{1:n-1}; \boldsymbol{\theta})
$$

By applying the Kalman filter, we can show that $p(\boldsymbol{Y}_n|\boldsymbol{Y}_{1:n-1}; \boldsymbol{\theta})$, the one-step-ahead forecast distributions for the observations, is normal density with mean vector $\boldsymbol{\mu}_n = \mathrm{E}(\boldsymbol{Y}_n|\boldsymbol{Y}_{1:n-1})$ and variance-covariance matrix $\Sigma_n(\boldsymbol{Y}_n|\boldsymbol{Y}_{1:n-1})$ [19]. Therefore, the log-likelihood can be written as follows.

$$
\begin{aligned}
l(\boldsymbol{\theta}) &= \log L(\boldsymbol{\theta}) \\
&= \sum_{n=1}^{N} \log p(\boldsymbol{Y}_n | \boldsymbol{Y}_{1:n-1}; \boldsymbol{\theta}) \\
&= \sum_{n=1}^{N} (2\pi)^{-2/K} (\det \boldsymbol{\Sigma}_n)^{-1/2} \prod_{k=1}^{K} \frac{1}{Y_n^{(k)}} \exp\left[-\frac{1}{2} \boldsymbol{d}_n' \boldsymbol{\Sigma}_n^{-1} \boldsymbol{d}_n\right]
\end{aligned}
$$

where $\boldsymbol{d}_n \equiv \log \boldsymbol{Y}_n - \boldsymbol{\mu}_n$.

Although we adopt the Gaussian assumption to error terms, it is theoretically straightforward to alter a Gaussian distribution to other distribution or a linear to non-linear model because we employ the state space representation. However, we need to apply the particle filter [20] in that case. From a practical perspective, it is not feasible to apply the particle filter to our framework because it becomes difficult to estimate the high-dimensional parameters and extend the method to multivariate case. Though the Gaussian assumption may seem strong, the Kalman filter not only achieves fast computation by employing the framework of least-squares method but also satisfies a desirable property of the estimator, best linear unbiased estimator (BLUE). Moreover, the goal of our study is to detect anomalies quickly and precisely, not to build an accurate time-series generation model. Construction of a sophisticated generative model with non-Gaussian distributions may lead to accurate anomaly detection. However, employment of such a model sacrifices the speed of anomaly detection and its extension to multiple dimensions, which are important in this study.

Our anomaly detection rule is based on the ratio of log-likelihoods. As a matter of convenience, we call the above model OUR model. To compute the ratio of log-likelihoods, we additionally construct an over-dispersed model that we call the NULL model. As in OUR model, the NULL model is estimated by the Kalman filter, but independently from OUR model and with fixed (exogenous) parameters $\boldsymbol{\theta}_0$ such that its dispersion is larger than that of OUR model. We define the anomaly score $r_n$ as the ratio of log-likelihood in OUR model to that in the NULL model at each time step. We use log-likelihood instead of likelihood to avoid canceling significant digits. We regard an observation $\boldsymbol{Y_n}$ as an anomaly if the anomaly score exceeds a threshold $\underline{r}$. Therefore, our anomaly detection rule is summarized as follows.

$$\forall n, \quad r_n \equiv \frac{l_n(\boldsymbol{\theta})}{l_n(\boldsymbol{\theta}_0)} > \underline{r} \;\; \Rightarrow \;\; \boldsymbol{Y}_n \text{ is anomaly}$$

where $l_n$ indicates the log-likelihood for observation $\boldsymbol{Y}_n$ and computed by taking difference between $l_{1:n}$, the log-likelihood computed by using observations from $\boldsymbol{Y}_1$ to $\boldsymbol{Y}_n$, and $l_{1:n-1}$.

The idea on our anomaly detection rule is quite simple and illustrated in Fig 1 for the univariate case when $\underline{r} = 1$. Anomalies arise in the tail of the distribution in OUR model and the probabilities of such observations arising are expected to be higher for the over-dispersed NULL model. It corresponds to the condition in our anomaly detection rule that the log-likelihood in OUR model is greater than that in the NULL model. Note that the log-likelihoods take negative values in Fig 1 and the negative signs are canceled out in the ratio of log-likelihoods. Our method can be applied to both supervised and unsupervised learning. If we have a labeled data set, the anomaly score can be defined as the ratio of conditional probability distributions given the labels (anomaly or not). On the other hand, if we only have an unlabeled data set, we cannot obtain the conditional probability distribution given the anomalous states. Our idea reflects that we supplement this conditional probability distribution by constructing the NULL model. Previous studies also employ threshold rules [8, 21–23]. However, most of them apply thresholds on generative models directly. Our model applies the threshold on log-likelihood ratio rather than the generative model.

Under the Kalman filter setting, log-likelihood functions of OUR and NULL models follow multivariate log-normal distributions with parameters $(\boldsymbol{\mu}_n, \boldsymbol{\Sigma}_n)$ and $(\boldsymbol{\mu}_{0,n}, \boldsymbol{\Sigma}_{0,n})$, respectively. Therefore, the ratio of the two values does not follow Gaussian distribution. It is important to note that the estimates of $\boldsymbol{\mu}_n$ and $\boldsymbol{\mu}_{0,n}$ vary with time and $\widehat{\boldsymbol{\mu}}_n \neq \widehat{\boldsymbol{\mu}}_{0,n}$ in general. This is evident from the actual calculation of the Kalman filter. With the Kalman filter, state estimation proceeds alternately repeating the prediction and filtering steps. In the filtering step, flitered state

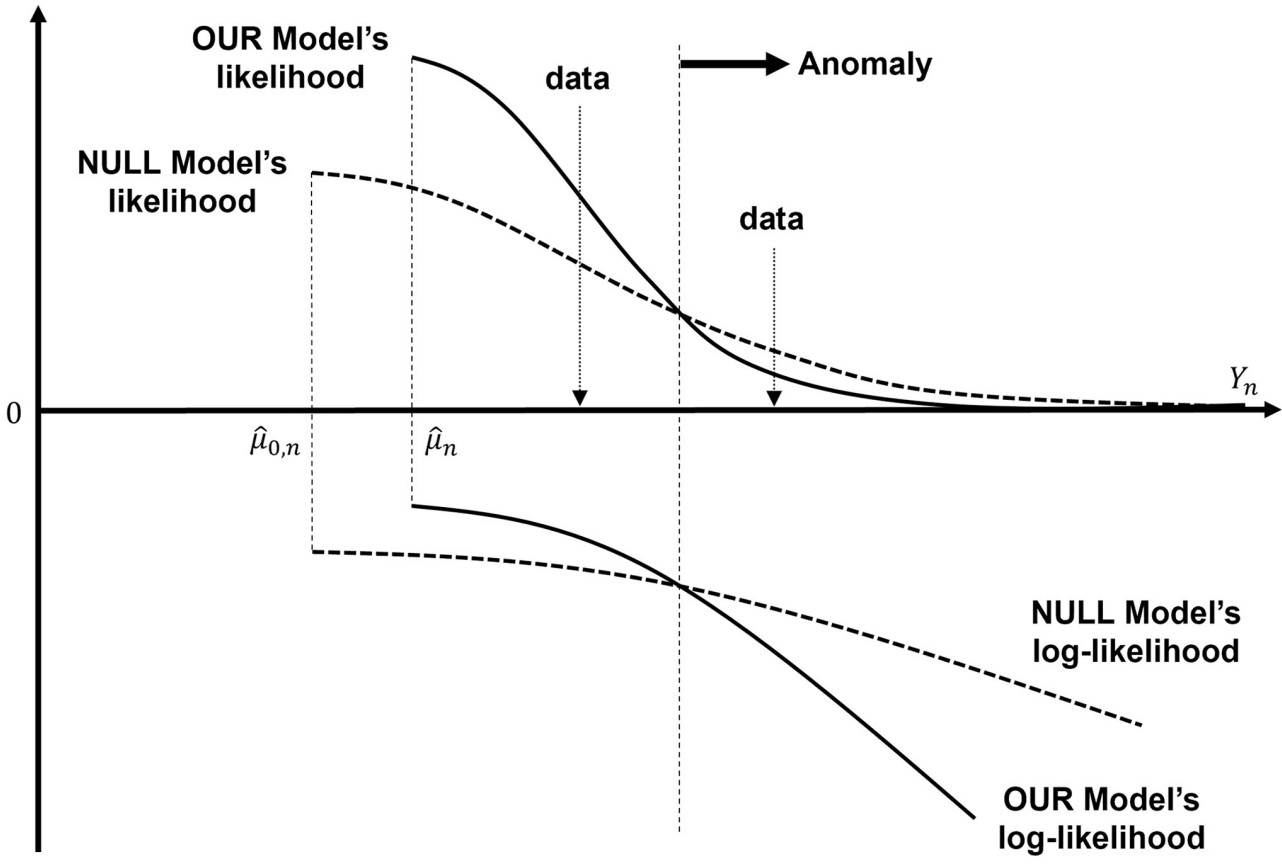

**Fig 1. Illustration of anomaly detection rule (Univariate case when $\underline{r} = 1$).**

estimator $\widehat{X}_{n|n}$ is obtained from a priori estimator $\widehat{X}_{n|n-1}$ as

$$
\begin{aligned}
\widehat{X}_{n|n} &= \widehat{X}_{n|n-1} + K_n(Y_n - F_n\widehat{X}_{n|n-1}), \\
K_n &= Q_{n|n-1}F_n'(F_nQ_{n|n-1}F_n' + V_n)^{-1}, \\
Q_{n|n} &= (I - K_nF_n)Q_{n|n-1}.
\end{aligned}
$$

$\widehat{X}_{n|n-1}$ and $Q_{n|n-1}$ are obtained by the prediction step as

$$
\begin{aligned}
\widehat{X}_{n|n-1} &= G_n\widehat{X}_{n-1|n-1}, \\
Q_{n|n-1} &= G_nQ_{n-1|n-1}G_n' + W_n.
\end{aligned}
$$

From the above equations, it is clear that $\widehat{\mu}_n$ and $\widehat{\mu}_{0,n}$ are obtained via $W_{n-1}$ and $V_n$, then $\widehat{\mu}_n \neq \widehat{\mu}_{0,n}$ in general. Furthermore, it is also clear that distribution of the ratio also varies with time step because of subscript $n$. Most of the previous papers that adopt the threshold rule assume Gaussian distributions for the error term explicitly or implicitly [21–23]. Our method applies the threshold rule to non-Gaussian and time varying distribution.

From a practical perspective, when we detect anomaly at time $n$, data $Y_n$ is replaced by NA. In other words, at time $n$, only the prediction step is done and the filtering step is skipped. If

this treatment is not done at $n$, anomaly data influences prediction state estimator $\widehat{X}_{n+1|n}$ via filtering in $n$. Then, ignorance of replacement makes it difficult to detect anomaly at $n + 1$ accurately. If our task is change point detection, this replacement is not needed. Our method uses Bayesian model, then it is easy to conduct this replacement.

In the next section, we apply an univariate version of the proposed method to labeled data sets for evaluation. We utilize two benchmark datasets: the Yahoo S5 data set (https://webscope.sandbox.yahoo.com/catalog.php?datatype=s&did=70) and the NAB data set (https://github.com/numenta/NAB). We utilize real data only from both benchmark. To collect the data, we comply with the terms and conditions for the websites. After that, we apply a bivariate version of the proposed method to unlabeled real-world data provided by Flat inc. (https://www.flat-inc.jp/). We describe further information of each data set in the evaluation and application section.

## Evaluation

In this section, we evaluate our anomaly detection method. Although we have unlabeled real-world data, we utilize labeled benchmark data sets to evaluate our method because a labeled data set is required to evaluate the method's performance.

### Data

The Yahoo S5 data set is a publicly available benchmark data set which is provided as part of the Yahoo! Webscope program [24]. It is widely used for evaluating anomaly detection methods [8, 21–23, 25, 26]. It consists of a total of 367 real and synthetic time series data with labeled anomalies. This data set is divided into four data sets namely the A1, A2, A3 and A4 benchmarks. The A1 benchmark contains real production traffic to some of the Yahoo! properties and the other three benchmarks contain synthetic time series. To evaluate our anomaly detection method, we utilize the A1 benchmark only. Because other three benchmarks also include the components such as trend and seasonality that generate the synthetic time series, we expect that our method can easily achieve high performance by including such components in our model specification. The A1 benchmark contains a total of 67 univariate time series. Actually, three time series have no true anomalies and there are five time series in which the test data includes no true anomalies when we divide the data into hold-out and test data. We describe how to split the data into hold-out and test data in the procedure subsection. Therefore, we exclude these eight time series from the evaluation and apply our method to a total of 59 A1 benchmark time series. The timestamps are hourly observed and the anomalies are marked by humans according to the description of the Yahoo S5 data set.

The NAB data set is a publicly available benchmark data set for anomaly detection, released by Numenta. It is also widely utilized for evaluating anomaly detection methods [8, 23, 25–27]. It consists of 58 labeled artificial and real time series data from various domains such as road traffic, network utilization and online advertisement. Out of 58 time series, we utilize six time series from online advertisement domain (realAdExchange) because the main focus of the study is on the Web related field. The realAdExchange data set includes real time series data on online advertisement clicking rates, where the metrics are cost-per-click (CPC) and cost per thousand impressions (CPM). Because one time series includes no true anomalies in the test data when we divide the data into hold-out and test data, we exclude it and utilize the remaining 5 time series for evaluation. The timestamps are hourly observed and the anomalies are marked as a result of the defined labeling procedure [28].

## Modeling

We consider an univariate DLM. In S1 Appendix, we show the state space representation of the model. The observation model is formulated as follows.

$$Y_n = T_n + S_n + H_n + A_n + \epsilon_{Y,n}, \quad \epsilon_{Y,n} \sim \mathcal{N}(0, \sigma_Y^2)$$

where $Y_n$ is the observation at time $n$ and $\epsilon_{Y,n}$ is the observation noise which follows a normal distribution with mean zero and variance $\sigma_Y^2$. As state variables, we consider the trend, 24-hour cycle (periodicity), hour and autoregressive (AR) components ($T_n$, $S_n$, $H_n$ and $A_n$, respectively). The system model is specified as follows.

$$
\begin{aligned}
T_n &= T_{n-1} + \omega_{T,n}, \quad \omega_{T,n} \sim \mathcal{N}(0, \sigma_T^2) \\
S_n &= -\sum_{j=1}^{23} S_{n-j} + \omega_{S,n}, \quad \omega_{S,n} \sim \mathcal{N}(0, \sigma_S^2) \\
H_n &= \sum_{k=1}^{23} h_{k,n} D_{k,n} \quad \text{where} \quad h_{k,n} = h_{k,n-1} + \omega_{H,k,n}, \quad \omega_{H,k,n} \sim \mathcal{N}(0, \sigma_H^2) \ \forall k \\
A_n &= \sum_{l=1}^{2} a_l A_{n-l} + \omega_{A,n}, \quad \omega_{A,n} \sim \mathcal{N}(0, \sigma_A^2)
\end{aligned}
$$

where $D_{k,n}$ in the hour component $H_n$ is the dummy variable for each hour $k$. For simplicity, we assume that the system noise $\omega_{H,k,n}$ has constant variance irrespective of hour $k$. Finally, we have the following seven parameters to estimate.

$$\boldsymbol{\theta} = (\sigma_Y^2, \sigma_T^2, \sigma_S^2, \sigma_H^2, \sigma_A^2, a_1, a_2)$$

## Procedure

Throughout the evaluation, we fix the model specification and apply the same model described in the last modeling subsection to all time series in the Yahoo A1 benchmark and the NAB realAdExchange data set. However, for each time series, we consider a total of four cases in terms of (a) the transformation of $Y_n$ and (b) the handling of detected anomalies. First, we consider whether or not to take the logarithm of $Y_n$ ("non-log" or "log" case). Second, as we mentioned in the method section, we consider whether or not to remove observations detected as anomalies in the future state estimation ("non-NA" or "NA" case). In practice, we sequentially perform anomaly detection. In other words, when we get a new observation $Y_n$ at time $n$, we decide whether or not $Y_n$ is anomalous by computing $l_n$ as the difference between $l_{1:n}$ and $l_{1:n-1}$. We repeat this process up to time $N$. Suppose that $Y_n$ is detected as anomalies and $Y_{n+1}$ is actually a normal observation. If we include $Y_n$ in the state estimation at time $n + 1$, $Y_{n+1}$ may likely be detected as an anomaly due to the sharp fluctuation of state variables estimated by the Kalman filter. In order to take into account the possibility of this phenomenon, we consider whether to remove observations detected as anomalies before time $n$ in the state estimation at time $n$. In the "NA" case, we treat observations detected as anomalies before time $n$ as NA in the state estimation at time $n$. Based on the combination of (a) and (b), we consider a total of four cases for each time series: (1) non-log and non-NA, (2) non-log and NA, (3) log and non-NA, and (4) log and NA.

Given time series $d$ and case (i), we perform the following procedure: First, we estimate the model's parameters $\boldsymbol{\theta}$ using the hold-out data. We divide the time series $d$ into hold-out and test data where the first one-third (33%) of the observations is used as hold-out data and the remaining (67%) observations are used as test data. In addition, we remove true anomalies in

hold-out data when estimating parameters because the estimated parameters may be biased if we estimate parameters using the hold-out data including anomalies. In practice, we treat true anomalies as NA when estimating the parameters using the hold-out data. We employ the maximum likelihood estimation (MLE) method for parameter estimation. For parameter search, we utilize the simulated annealing (SA) algorithm. We try several initial values of parameters and choose the estimates that provide the largest likelihood to avoid converging to local solutions. Next, given the estimated parameters $\widehat{\boldsymbol{\theta}}$ for OUR model and $\boldsymbol{\theta}_0$ for NULL model, we perform state estimation of OUR and NULL model by Kalman filter using the test data and perform anomaly detection based on our anomaly detection rule described in the method section. Specifically, at time $n$, we independently perform state estimation of OUR and NULL model by Kalman filter. Then, we obtain the value of $l_n(\widehat{\boldsymbol{\theta}})$ and $l_n(\boldsymbol{\theta}_0)$ by computing $l_{1:n}(\widehat{\boldsymbol{\theta}}) - l_{1:n-1}(\widehat{\boldsymbol{\theta}})$ and $l_{1:n}(\boldsymbol{\theta}_0) - l_{1:n-1}(\boldsymbol{\theta}_0)$, respectively. After that, we determine if the observation $Y_n$ is anomalous by comparing its ratio, defined as $r_n$, with the threshold $\underline{r}$. We repeat this process up to time $N$.

We evaluate the performance of our method using $F_1$ score. This is the most commonly used evaluation metric in the literature on anomaly detection and defined as follows:

$$F_1 = 2 \cdot \frac{\text{Precision} \cdot \text{Recall}}{\text{Precision} + \text{Recall}}$$

where $\text{Precision} = \frac{TP}{TP+FP}$ and $\text{Recall} = \frac{TP}{TP+FN}$. In these equations, TP (True-Positive) is the number of anomalies that are correctly detected as anomalies, FP (False-Positive) is the number of non-anomalies (normal observations) that are detected as anomalies, and FN (False-Negative) is the number of anomalies that are not detected as anomalies. In other words, precision is the ratio of true anomalies over detected anomalies and recall is the ratio of correctly detected anomalies over true anomalies. The $F_1$ score is defined as the harmonic mean between precision and recall.

Note that there are two exogenous parameters in our method. One is the parameters in the NULL model and the other is the threshold $\underline{r}$ in the anomaly detection rule. In terms of the parameters in the NULL model, we set $a_1$ and $a_2$ to 0 and the variance parameters to $k$ times the estimated variance parameters in OUR model such that the dispersion of the NULL model is greater than that of OUR model. We consider $k \in \{10^1, 10^{1.5}, 10^2, \cdots, 10^{6.5}, 10^7\}$ for "non-log" case (case (1) and (2)) and $k \in \{e^1, e^{1.5}, e^2, \cdots, e^{4.5}, e^5\}$ for "log" case (case (3) and (4)). For each value of $k$, we compute the optimal threshold $\underline{r}_k^*$ based on the Receiver Operating Characteristics (ROC) curve such that the $F_1$ score is maximized, which is a similar approach as Munir et al. [8] and Thill et al. [22]. Under each pair of $k$ and $\underline{r}_k^*$, we perform anomaly detection using the test data and choose the highest $F_1$ score as the performance of our method on time series $d$ and for case (i), say $F_1^{(d,i)}$. In summary, we can obtain four $F_1$ scores for time series $d$, $F_1^{(d,1)}$, $F_1^{(d,2)}$, $F_1^{(d,3)}$ and $F_1^{(d,4)}$. Then, we choose the highest one as the performance of our method on time series $d$.

## Evaluation results

**Results on Yahoo A1 benchmark.** In Table 1, we show our method's average $F_1$ score, precision, and recall across 59 time series in the A1 benchmark data set. In S1 Table, we show the optimal threshold, $F_1$ score, precision, and recall on each of 59 time series in the A1 benchmark under each case. For comparison, we also show the best average scores which have been reported in five other academic papers [8, 21–23, 25]. Each of them compares the performance of several anomaly detection methods utilizing the Yahoo A1 benchmark. Munir et al. [8]

**Table 1. Performance on Yahoo A1 benchmark.**

|  | $F_1$ | Precision | Recall |
|---|---|---|---|
| **Our method** | **0.73** | **0.75** | **0.77** |
| Munir et al. [8] | 0.48 | - | - |
| Suh et al. [21] | 0.52 | 0.54 | 0.51 |
| Thill et al. [22] | 0.67 | 0.66 | 0.67 |
| Däubener et al. [25] | 0.58 | 0.48 | 0.72 |
| Maciąg et al. [23] | 0.70 | 0.66 | 0.79 |

proposes a novel deep learning-based anomaly detection approach (DeepAnT) and performs an evaluation of 15 algorithms on 10 anomaly detection benchmarks including the Yahoo A1 benchmark. They report that the highest average $F_1$ score on the Yahoo A1 benchmark is 0.48 which is achieved by Twitter's anomaly detection method, commonly referred to as AdVec. On the other hand, their proposed method, DeepAnT, achieves an average $F_1$ score of 0.46. Suh et al. [21] presents an echo-state conditional variational autoencoder (ES-CVAE) and evaluate it using an 11 time series in the A1 benchmark. They report of its average $F_1$ score is 0.52. Thill et al. [22] presents a comparative study on several online anomaly detection algorithms using the Yahoo S5 data set. They report that the Simple Online Regression Anomaly Detector (SORAD) is quite successful and its average $F_1$ score on the Yahoo A1 benchmark is 0.67. Däubener et al. [25] empirically compare the common machine learning and statistical methods in terms of detecting anomalies. They report that Gaussian processes and a support vector machine achieve slightly better performances than other algorithms and the average $F_1$ score of both algorithms on the Yahoo A1 benchmark is 0.58. The scores listed in Table 1 are achieved by Gaussian processes. For your information, the $F_1$ score, precision, and recall of the support vector machine are 0.58, 0.46, and 0.76, respectively. Maciąg et al. [23] adapts the Online evolving Spiking Neural Network (OeSNN) classifier to the anomaly detection task and proposes an Online evolving Spiking Neural Network for Unsupervised Anomaly Detection algorithm (OeSNN-UAD). Comparing the proposed method with state-of-the-art anomaly detection methods using the Yahoo A1 benchmark, their method achieves the highest average $F_1$ score of 0.70. Table 1 shows that our method achieves the average $F_1$ score of 0.73 and outperforms these existing methods.

Why can our simple method outperform other complex schemes? The reasons for this are twofold. First, our method takes into account the specific information that captures the feature of time series data at the modeling stage, e.g. what time it is. In other words, the number of explanatory variables in the model is much larger in our method than others because other methods basically utilize past observations only when predicting future observations. For comparison, we apply our method including the trend component (random walk) only, the simplest model that is often called a local level model, to the Yahoo A1 benchmark. In S2 Table, we show the optimal threshold, $F_1$ score, precision, and recall on each of 59 time series in the A1 benchmark under each case. The average $F_1$ score is 0.63. This is comparable to the existing methods' performance, but is lower than our method's performance, 0.73 in Table 1. It indicates that including the time-series-specific information such as hour component into the model contributes to improving the performance.

Second, the property of the data might affect the performance of the method. Braei and Wagner [26] perform a comparative evaluation of 20 anomaly detection methods from statistical, machine learning and deep learning methods using Yahoo S5 data set and one of the NAB data set, the data on New York City taxi demand. Their results show that (1) the statistical

models achieve better results while the deep learning methods generally perform poorly, and most of the machine learning approaches are located in the center for Yahoo S5 data set, which is the similar finding to ours, and (2) the deep learning methods perform much better on the NAB data set while the statistical approaches achieve a very low performance. They claim that the statistical models perform better for Yahoo S5 data set because the anomalies in this data set are either point or collective anomalies while NAB data set contains contextual anomalies. Therefore, they finally conclude that the property of the data affects the performance of the algorithms. Although the high performance of our method might also stem from the property of the data, we can at least argue that our method performs well on detecting point or collective anomalies.

In Fig 2, we show the best $F_1$ score on all time series in the Yahoo A1 benchmark. The time series with no $F_1$ score are removed ones from the evaluation. We can see that our method achieves relatively good performance on most of the time series: $F_1$ score is greater than 0.5 on 88% of the time series. For example, we show the anomaly detection result on time series No.36 in Fig 3. The dotted line, squares, x-marks and solid vertical line indicate observations, true anomalies, detected anomalies and the border line between hold-out and test data, respectively. The $F_1$ score is 1.00, achieved under the case (2), and we can see that our method perfectly detect the true anomaly in the test data. In addition, we show the estimates of the state variables for time series No.36 in Fig 4. In the top figure in Fig 4, the dashed line indicates the observations and the solid line indicates the trend estimates. We can see that each component actually contributes to the prediction, which supports the aforementioned reason why our simple method outperforms other complex schemes.

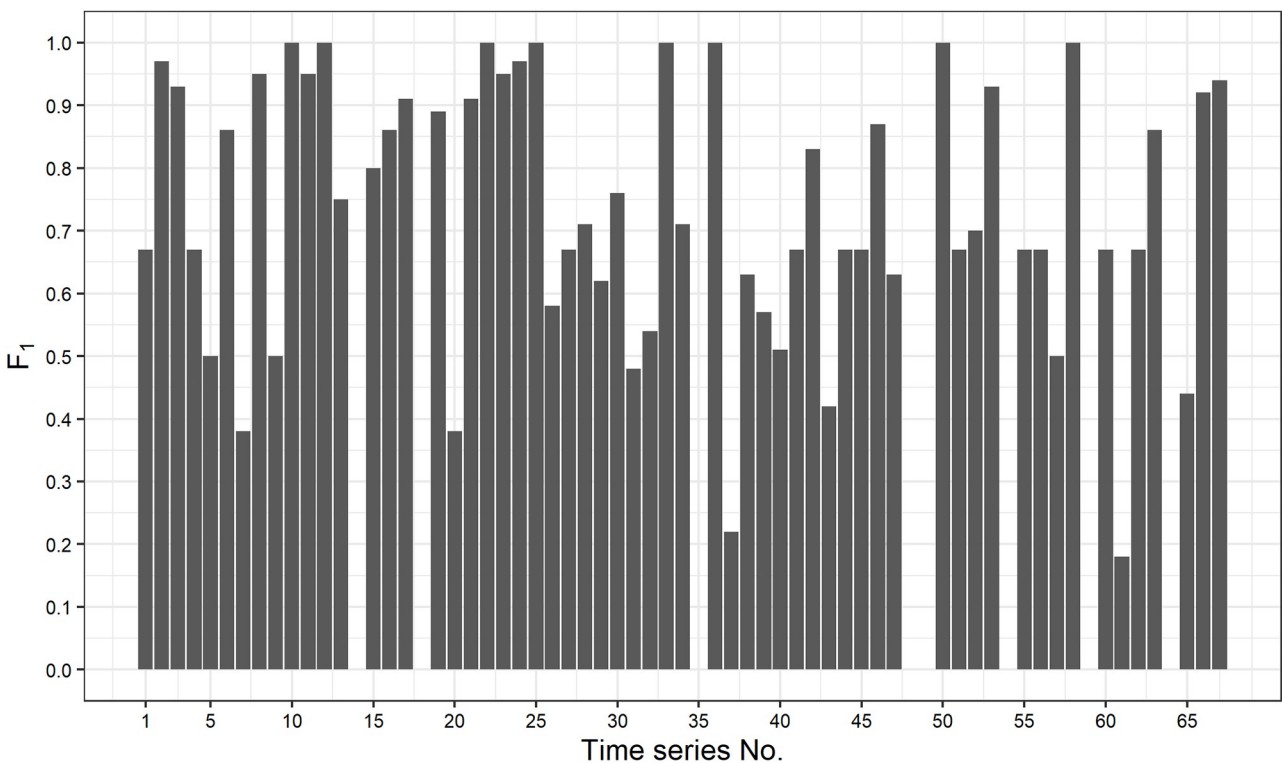

**Fig 2. Best $F_1$ scores on all A1 benchmark time series.**

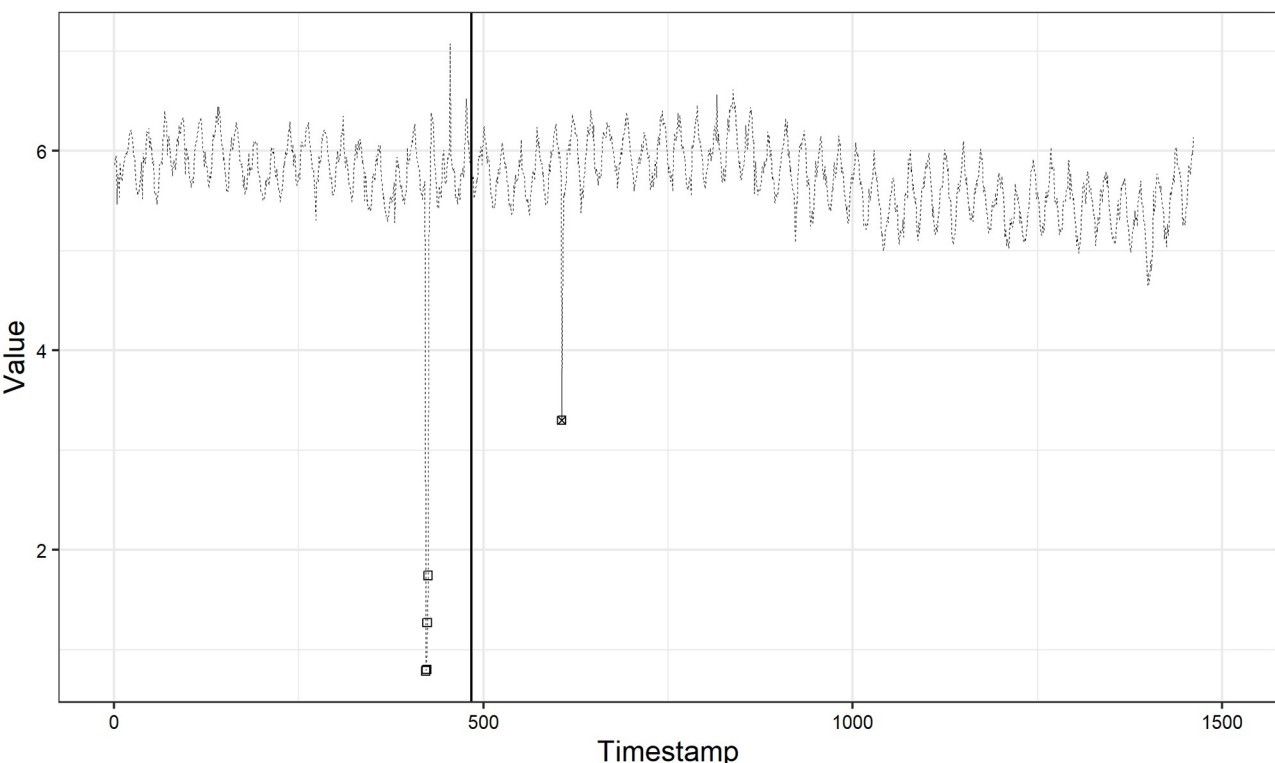

**Fig 3. Anomaly detection result on time series No.36.**

On the other hand, there exist some time series with quite low $F_1$ scores, even though we choose the best score from among four cases. As an example, we show the anomaly detection result on time series No.61 in Fig 5. The $F_1$ score is 0.18. This is the worst score among the 59 time series, while this is the best score for time series No.61 achieved under the case (3). We can see that there are many false positives, i.e., our method detects many normal observations as anomalies. Actually, the precision is quite low, at 0.11, while the recall is relatively large, at 0.65.

What causes low scores? First, the model might be misspecified. The low score, in particular, low precision suggests that the model fails to learn normal behavior of the data. In this evaluation, we fix the model specification and apply the same model to all 59 A1 benchmark time series. The failure of learning is likely to stem from the lack of appropriate modeling for individual time series. We need additional information to more accurately learn the normal behavior of individual time series. However, it is noteworthy that our method achieves relatively high scores on most of the time series in the A1 benchmark data set, even under the fixed model specification, with little information on the data.

Second, our method might be bad at change point detection. The time series No.61 looks like a time series suitable for applying a change point detection method rather than an anomaly detection method: it seems that a change has occurred at around the timestamp of 600 in Fig 5. In addition, we observe that the behaviors of most low-scoring time series are similar to the time series No.61 and might be suitable for applying the change point detection method. Nonetheless, our method is potentially capable of detecting a change point by virtue of the treatment of whether or not we remove the observations detected as anomalies in future state estimation, as described in the last subsection. Actually, in Fig 5, our method can detect a lot of true anomalies at around the timestamp of 600 and a true anomaly at around the timestamp of 1500, which results in a relatively large recall of 0.65.

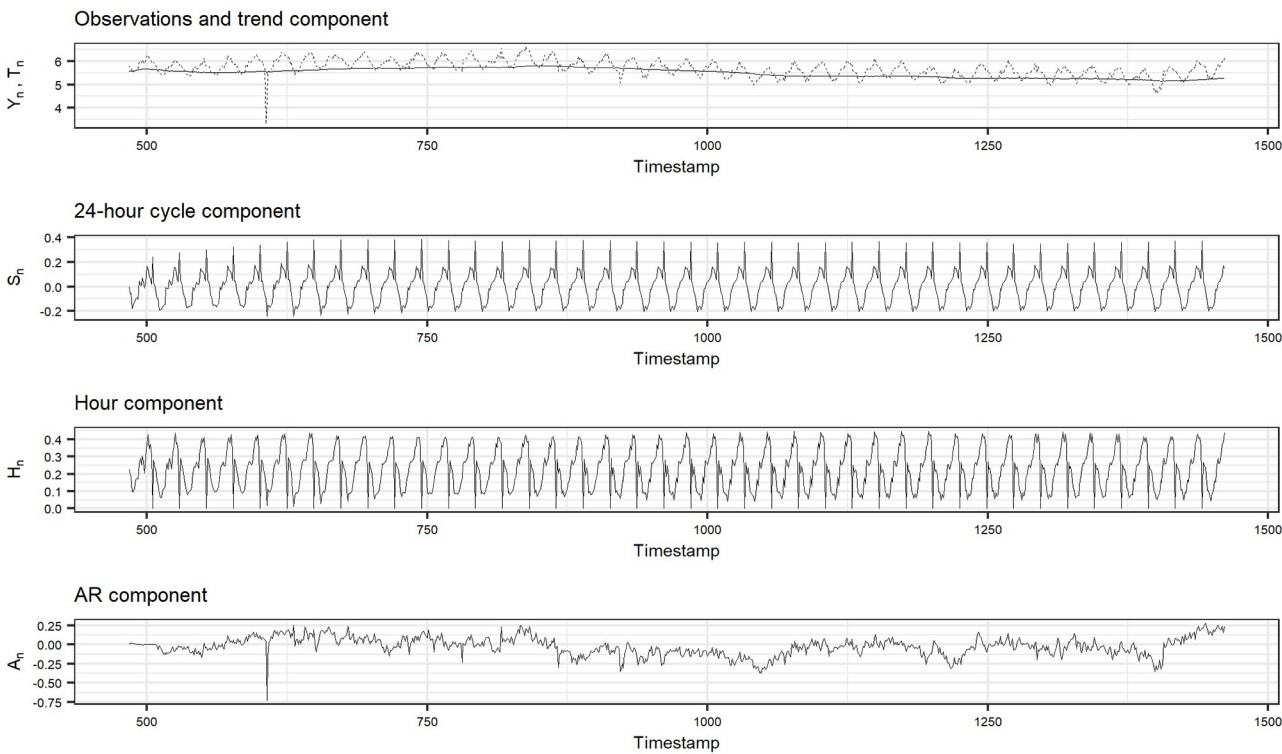

**Fig 4. Estimates of the state variables for time series No.36.**

**Results on NAB realAdExchange.** In Table 2, we show our method's average $F_1$ score, precision, and recall across 5 time series in the realAdExchange data set. In S3 Table, we show the optimal threshold, $F_1$ score, precision, and recall on each of 5 time series in the realAdExchange data set under each case. As in Table 1, we also show the best average scores which have been reported in four other academic papers [8, 23, 25, 27] in Table 2. Three studies except for Amarbayasgalan et al. [27] utilize not only Yahoo A1 benchmark but also realAdExchange data set to compare the performance of several anomaly detection methods. First, Munir et al. [8] reports that their proposed method, DeepAnT, achieves the highest $F_1$ score of 0.13 on the realAdExchange data set. Second, Däubener et al. [25] reports that the Twitter's anomaly detection method, AdVec, achieves the highest $F_1$ score of 0.50 on the realAdExchange data set. Third, Maciąg et al. [23] reports that their proposed method, OeSNN-UAD, achieves the highest $F_1$ score of 0.23 on the realAdExchange data set. Finally, Amarbayasgalan et al. [27] proposes a deep learning-based unsupervised anomaly detection approach for time series data (RE-ADTS) and compares the proposed method with 10 state-of-the-art anomaly detection methods using NAB data set. They find that their proposed RE-ADTS outperforms the state-of-the-art methods and report that the average $F_1$ score of RE-ADTS on the realAdExchange data set is 0.23. Table 2 shows that our method achieves better or comparable performance compared to these existing methods.

We can also see that the Twitter's AdVec solely achieves the same $F_1$ score as our method, which is reported by Däubener et al. [25]. It performs a seasonal-trend decomposition based on Loess regression and applies the generalized Extreme Studentized Deviate (ESD) test using the obtained residuals to detect anomalies. Although AdVec and our method do something similar to detect anomalies, our method can additionally take into account the other variables

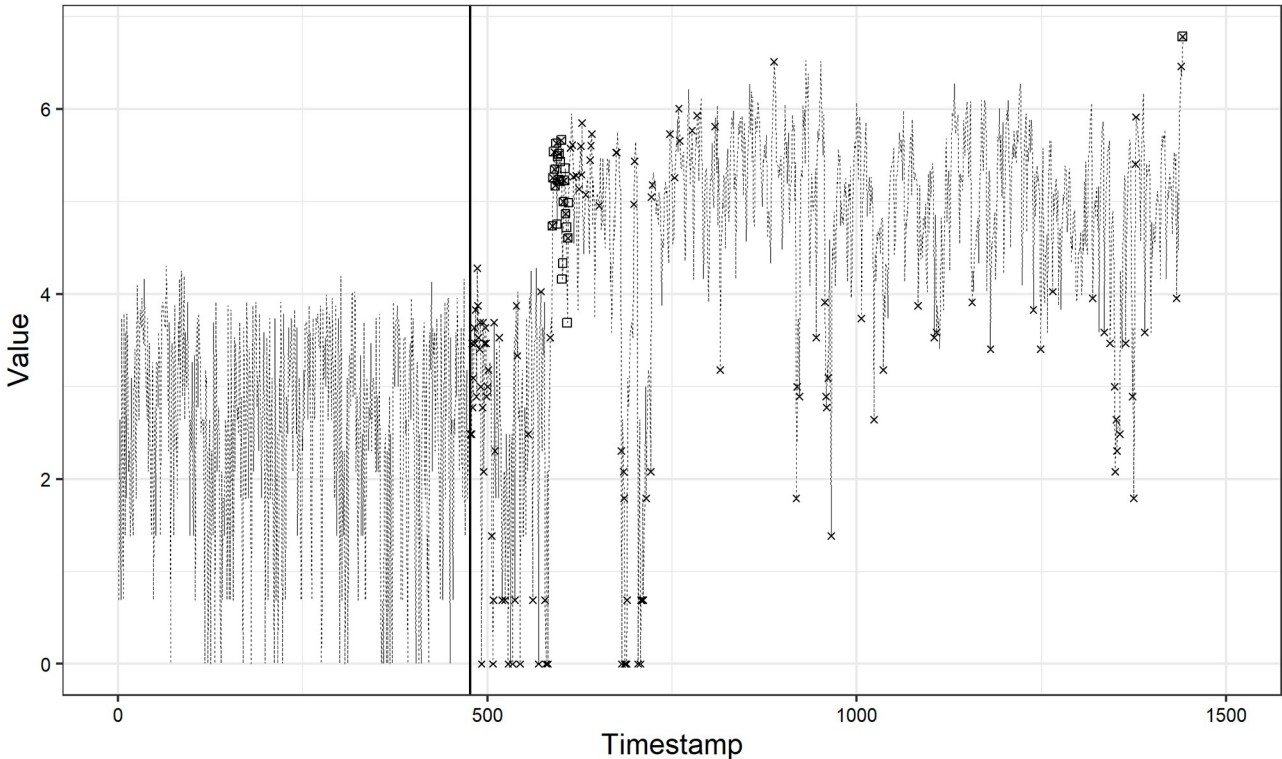

**Fig 5. Anomaly detection result on time series No.61.**

such as AR processes for predicting future observations. On the other hand, the common feature of AdVec and our method is that both methods take into account the specific information on time series data such as trend and seasonality in the model. Note that they achieve the highest $F_1$ score in Table 2. This result may stem from the fact that they utilize not only the past observations but also the specific information on time series data such as trend and seasonality, which is also discussed in the Yahoo A1 benchmark results. Our evaluation results imply that it is essential in time series anomaly detection to incorporate the specific information on time series data into the model.

In Figs 6 and 7, we show the best and worst anomaly detection result on realAdExchange data set, respectively. The $F_1$ score is 0.80 achieved under the case (1) in Fig 6 and 0.29 achieved under the case (2) in Fig 7, respectively. In Fig 6, the $F_1$ score is quite high and we can see that our method can detect two large spikes while it misses the remaining one true anomaly in the test data. On the other hand, in Fig 7, we can observe some false positives resulting in the low precision of 0.17 although our method accurately detect the true anomaly in the test

**Table 2. Performance on NAB realAdExchange.**

|  | $F_1$ | Precision | Recall |
|---|---|---|---|
| **Our method** | **0.50** | **0.45** | **0.73** |
| Munir et al. [8] | 0.13 | - | - |
| Däubener et al. [25] | 0.50 | 0.38 | 0.71 |
| Maciąg et al. [23] | 0.23 | 0.22 | 0.26 |
| Amarbayasgalan et al. [27] | 0.23 | 0.30 | 0.19 |

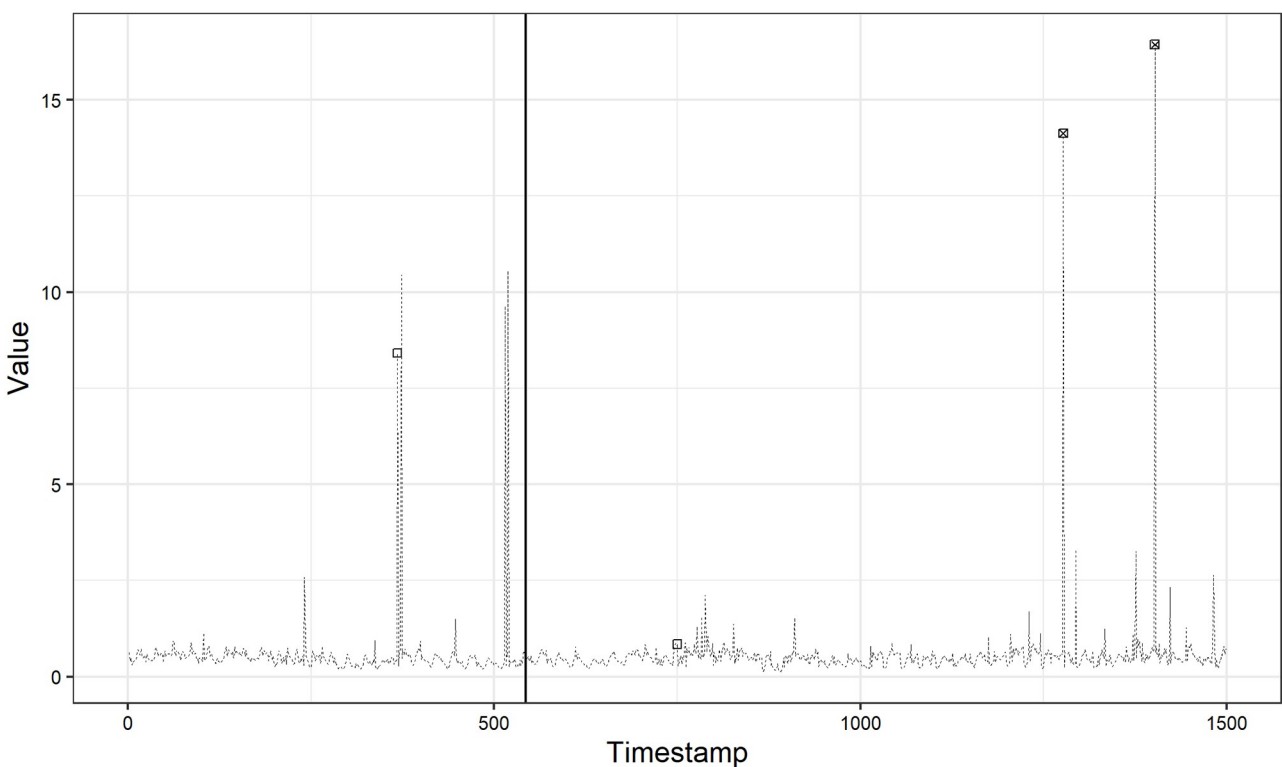

**Fig 6. Anomaly detection result on exchange-4_cpm_results.**

data. One of the reason for this observation is a misspecification of the model as we mentioned in the Yahoo A1 benchmark results. Another reason may include the anomaly labels marked as a result of the defined labeling procedure [28]. Munir et al. [8], Däubener et al. [25] and Maciąg et al. [23] also point out the same issue.

In Fig 7, our method detects two large spikes in the test data as anomalies, but these two observations are not labeled as anomalies. With the information on the data at hand alone, it is unclear why these two large spikes are not labeled as anomalies. If these observations were labeled as anomalies, our method's performance would become higher because of the increase in precision. In addition, with additional information, our method may not detect these observations as anomalies in the first place. The same applies to Fig 6. In Fig 6, our method misses one true anomaly in the test data. Without additional information, it is also unclear why this observation is labeled as anomaly. Because it is not understandable why the observation is (not) labeled as anomaly by observing the provided univariate time series only, we can expect that these univatiate time series may originally be a part of multivariate time series data. From Tables 1 and 2, we can see that the scores on NAB realAdExchange data set are totally lower than on Yahoo A1 benchmark data set. This observation could stem from the NAB's labeling procedure.

## Application

In this section, we apply our anomaly detection method to Web time series data to show our method's applicability to unlabeled real-world data. Because it would be a common situation where one must monitor multiple variables simultaneously, we now consider a bivariate version of our method.

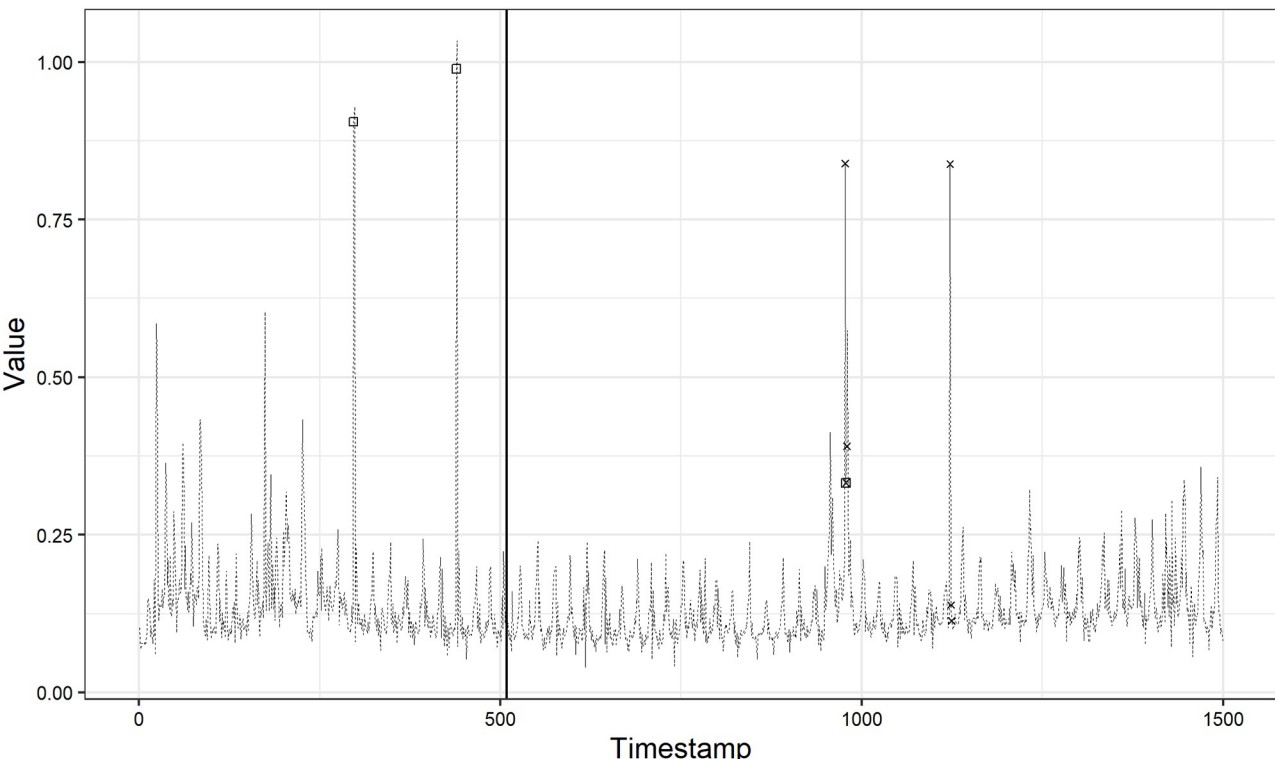

**Fig 7. Anomaly detection result on exchange-3_cpc_results.**

## Data

We use daily PV and SD data on an EC site that deals in insurance goods. The data period is from January 15, 2015 to January 14, 2018 and the sample size is 1,096. We show the data on PV and SD in Fig 8. Besides the data on daily PV and SD, we have four additional variables in terms of daily expenditure on digital advertising, which falls under the jurisdiction of the web advertising agency, namely two variables for expenditure on display advertising (the Google Display Network and Yahoo Display Ad Network) and two variables for that on listing advertising (Google and Yahoo). We utilize the first 365 observations from the study period as hold-out data for estimating the parameters of the model and the remaining 731 observations as test data for anomaly detection.

## Modeling

We construct a bivariate DLM based on the information available for a web advertising agency in the context of the introduction section. In S1 Appendix, we show the state space representation of the model. A Web time series data generally have the following characteristics: (1) there exists a trend and a periodicity such as day-of-week periodicity, (2) KPIs such as PV vary depending on weekdays or holidays, and (3) the impact of advertising on KPIs is not negligible. Therefore, we incorporate these factors in our model. The observation model is formulated as follows. Note that the following model represents the observation model for PV when $i = 1$ and for SD when $i = 2$.

$$Y_{i,n} = T_{i,n} + D_{i,n} + H_{i,n} + S_{i,n} + P_{i,n} + A_{i,n} + \epsilon_{Y_i,n}, \quad \epsilon_{Y_i,n} \sim \mathcal{N}(0, \sigma^2_{Y_i}), \quad i = 1, 2$$

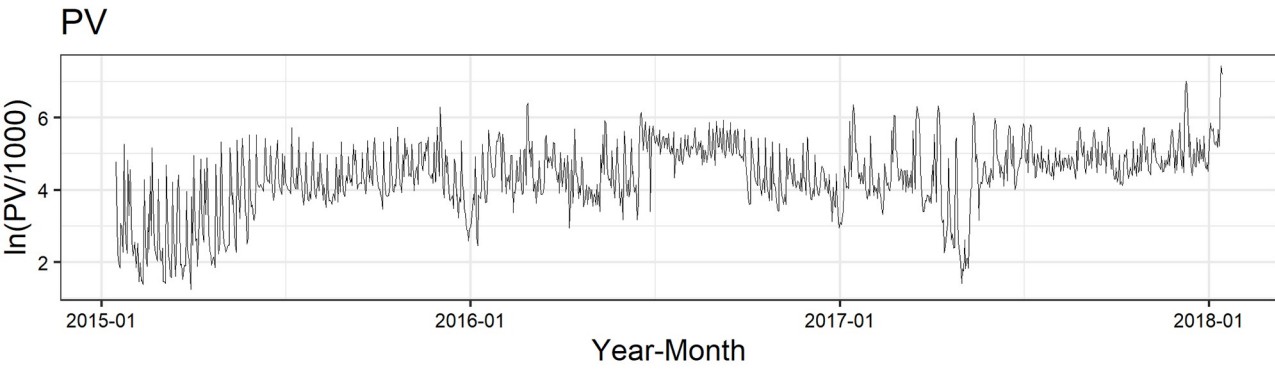

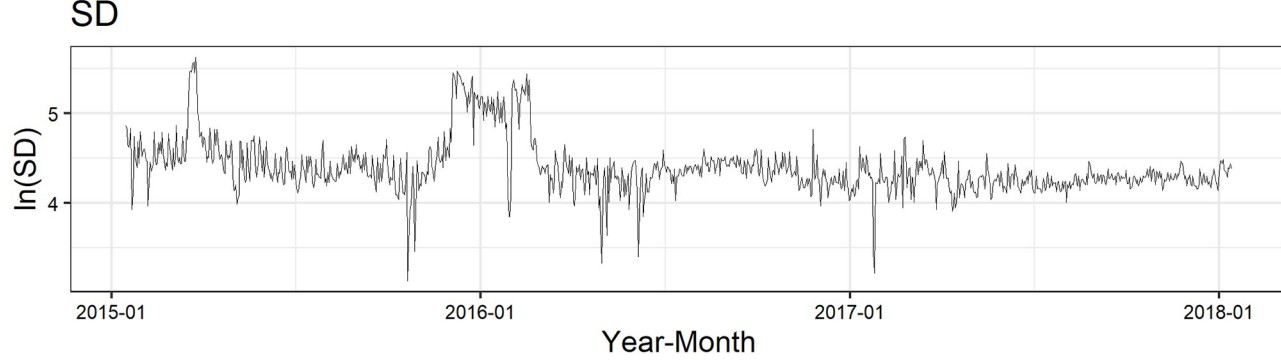

**Fig 8. Data for application.**

where $Y_{i,n}$ is the observation of time series $i$ on day $n$ and $\epsilon_{Y_i,n}$ is the observation noise for time series $i$ on day $n$. We assume that the observation noises $\epsilon_{Y_1,n}$ and $\epsilon_{Y_2,n}$ correlate with each other and specify the covariance as $\sigma^2_{Y_1 Y_2}$. Other variables are state variables. As state variables, we first consider the trend component $T_{i,n}$ that captures the longitudinal trend of time series $i$. Second, we include the day-of-week, holiday, season, and advertising components ($D_{i,n}$, $H_{i,n}$, $S_{i,n}$, and $P_{i,n}$, respectively) to control their effects on time series $i$. Finally, we include the AR component $A_{i,n}$ to capture the middle-term periodicity of time series $i$. The system model is specified as follows. For $i = 1, 2$,

$$T_{i,n} = T_{i,n-1} + \omega_{T_i,n}, \quad \omega_{T_i,n} \sim \mathcal{N}(0, \sigma^2_{T_i})$$

$$D_{i,n} = -\sum_{j=1}^{6} D_{i,n-j} + \omega_{D_i,n}, \quad \omega_{D_i,n} \sim \mathcal{N}(0, \sigma^2_{D_i})$$

$$H_{i,n} = H_{i,n-1} + \omega_{H_i,n}, \quad \omega_{H_i,n} \sim \mathcal{N}(0, \sigma^2_{H_i})$$

$$S_{i,n} = \sum_{k=1}^{3} s_{i,k,n} E_{k,n} \quad \text{where} \quad s_{i,k,n} = s_{i,k,n-1} + \omega_{S_i,k,n}, \quad \omega_{S_i,k,n} \sim \mathcal{N}(0, \sigma^2_{S_i,k}) \ \forall k$$

$$P_{i,n} = \sum_{l=1}^{4} p_{i,l,n} Z_{l,n} \quad \text{where} \quad p_{i,l,n} = p_{i,l,n-1} + \omega_{P_i,l,n}, \quad \omega_{P_i,l,n} \sim \mathcal{N}(0, \sigma^2_{P_i,l}) \ \forall l$$

$$A_{i,n} = \sum_{m=1}^{2} a_{i,m} A_{i,n-m} + \omega_{A_i,n}, \quad \omega_{A_i,n} \sim \mathcal{N}(0, \sigma^2_{A_i})$$

where $\omega_{\cdot,n}$ in each component is the system noise which we assume have no correlations with

each other. $E_{k,n}$ in the season component $S_{i,n}$ is a dummy variable. For example, $E_{1,n}$ takes 1 if day $n$ is in Spring and 0 otherwise. $Z_{l,n}$ in the advertising component $P_{i,n}$ is the advertising expenditure on day $n$. For example, $Z_{1,n}$ is the advertising expenditure on the Google Display Network on day $n$. We have a total of 29 parameters to estimate and they are summarized as follows.

$$\boldsymbol{\theta} = (\sigma^2_{Y_i}, \sigma_{Y_1 Y_2}, \sigma^2_{T_i}, \sigma^2_{D_i}, \sigma^2_{H_i}, \sigma^2_{S_i,1}, \sigma^2_{S_i,2}, \sigma^2_{S_i,3}, \sigma^2_{P_i,1}, \sigma^2_{P_i,2}, \sigma^2_{P_i,3}, \sigma^2_{P_i,4}, \sigma^2_{A_i}, a_{i,1}, a_{i,2})$$
$$\text{for} \quad i = 1, 2$$

## Pre-processing

Before estimating parameters $\boldsymbol{\theta}$ using hold-out data, we first remove what appears to be anomalies in the hold-out data. This is because (i) the hold-out data is expected to include anomalies and (ii) the estimated parameters may be biased if we estimate parameters with the hold-out data including anomalies. In this pre-processing, we first estimate the model's parameters through the MLE method. As in the evaluation section, we employed the SA algorithm for the parameter search. Next, given the estimated parameters, we apply the Kalman filter and obtain the smoothing values for PV and SD. We regard an observation in the hold-out data as an anomaly if the residual, defined as the absolute difference between the observed and smoothed value, is greater than a particular threshold. Because our data is unlabeled, the number of anomalies is preliminarily unknown. As a reference, we use the anomalous rate of the Yahoo A1 benchmark, the real Web time series data which we utilize in the evaluation section. The average anomalous rate across the 64 Yahoo A1 benchmark data sets is about 1.8%. Here, we exclude three data sets including no true anomalies. Therefore, we assume that 1.8% of the observations are anomalies in our data and set the threshold value to 1.6 for PV and 0.2 for SD such that the anomalous rate is 1.8%. Based on this criterion, we removed 7 observations as anomalies from the hold-out data for each of PV and SD.

## Parameter estimation

After removing anomalies in the hold-out data, we estimate the model's parameters $\boldsymbol{\theta}$ again by using the anomaly-removed hold-out data. As in the pre-processing, we employ the MLE method to estimate parameters and the SA algorithm for the parameter search. Table 3 shows the estimated parameters $\widehat{\boldsymbol{\theta}}$. We can see that most of the estimated variance parameters are small, all of the estimated variance parameters are less than 1.

As described in the evaluation section, we have two exogenous parameters in our method. One is the parameters in the NULL model and the other is the threshold $\underline{r}$ in the anomaly

**Table 3. Parameter estimates.**

| Parameter | Estimate | Parameter | Estimate | Parameter | Estimate | Parameter | Estimate |
|---|---|---|---|---|---|---|---|
| $\sigma^2_{Y_1}$ | 0.2921 | $\sigma^2_{P_1,2}$ | 0.0801 | $\sigma^2_{Y_2}$ | 0.0065 | $\sigma^2_{P_2,2}$ | 0.5958 |
| $\sigma^2_{T_1}$ | 0.0223 | $\sigma^2_{P_1,3}$ | 0.0030 | $\sigma^2_{T_2}$ | 0.0022 | $\sigma^2_{P_2,3}$ | 0.0284 |
| $\sigma^2_{D_1}$ | 0.0031 | $\sigma^2_{P_1,4}$ | 0.0032 | $\sigma^2_{D_2}$ | 0.0001 | $\sigma^2_{P_2,4}$ | 0.0023 |
| $\sigma^2_{H_1}$ | 0.0038 | $\sigma^2_{A_1}$ | 0.0044 | $\sigma^2_{H_2}$ | 0.0049 | $\sigma^2_{A_2}$ | 0.0126 |
| $\sigma^2_{S_1,1}$ | 0.0014 | $a_{1,1}$ | -0.3905 | $\sigma^2_{S_2,1}$ | 0.0051 | $a_{2,1}$ | 0.7513 |
| $\sigma^2_{S_1,2}$ | 0.0424 | $a_{1,2}$ | -0.0366 | $\sigma^2_{S_2,2}$ | 0.0471 | $a_{2,2}$ | -0.0074 |
| $\sigma^2_{S_1,3}$ | 0.0620 | $\sigma^2_{Y_1 Y_2}$ | 0.0321 | $\sigma^2_{S_2,3}$ | 0.2050 | - | - |
| $\sigma^2_{P_1,1}$ | 0.0016 | - | - | $\sigma^2_{P_2,1}$ | 0.0063 | - | - |

detection rule. Note that these two exogenous parameters play the same role in anomaly detection: a lower value increases the probability of making a Type I error (false positives) and a higher value increases the probability of making a Type II error (false negatives). Therefore, in this application, we fix the parameters in the NULL model such that the dispersion of NULL model is greater than that of OUR model. In terms of the threshold $\underline{r}$, we perform comparative statics in the next subsection and determine the value of the threshold based on it. Based on the parameter estimates in Table 3, we set all variance parameters to 1, covariance parameter to 0.1 and $a_{1,1} = a_{1,2} = a_{2,1} = a_{2,2} = 0$.

## Application results

In Fig 9, we show the application result when we set the threshold $\underline{r}$ to 3.0 under "NA" case. The solid-red and blue-dashed lines indicate one-step ahead forecast of observations for the OUR and NULL models, respectively. The detected anomalies are shown by x-marks. We can see that the variance of the NULL model is greater than that of OUR model, as expected, and a total of eight days are detected as anomalies. In the bivariate case, the likelihood function is defined as joint density, i.e., $L(\theta) = p(Y_{1,1}, \cdots, Y_{1,N}, Y_{2,1}, \cdots, Y_{2,N}; \theta)$. Therefore, in this result, it can be interpreted that our method regards a day as an anomaly if at least one of either PV or SD is anomalous on that day. Note that both PV and SD are marked with x-marks on the day that is detected as anomaly in Fig 9. In addition, the details of the detected anomalies are shown in Fig 10. We extract the observations within two weeks around the detected anomalous days from Fig 9. Focusing on the detected anomalies, we can observe that PV and SD tend to move in the opposite directions. We can also see that our method tends to detect the observations that deviated significantly from the one-step ahead forecast as anomalies. Because this

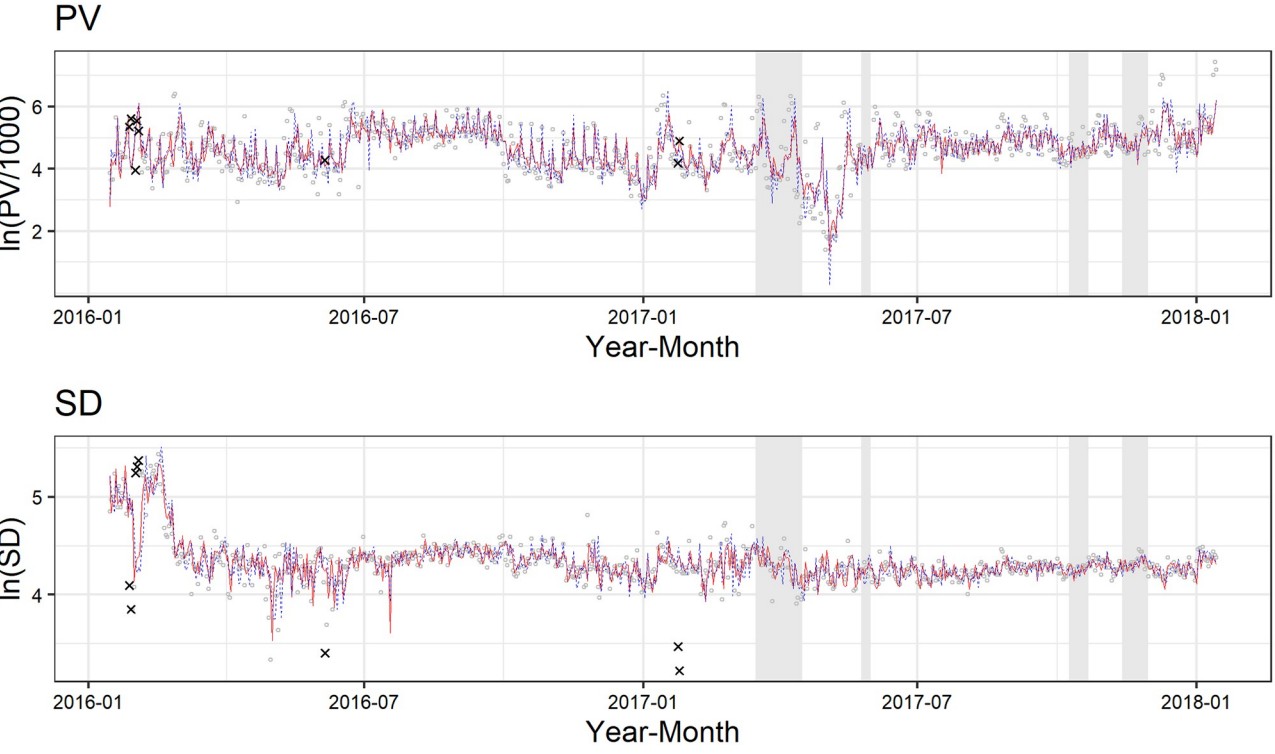

**Fig 9. Application result.**

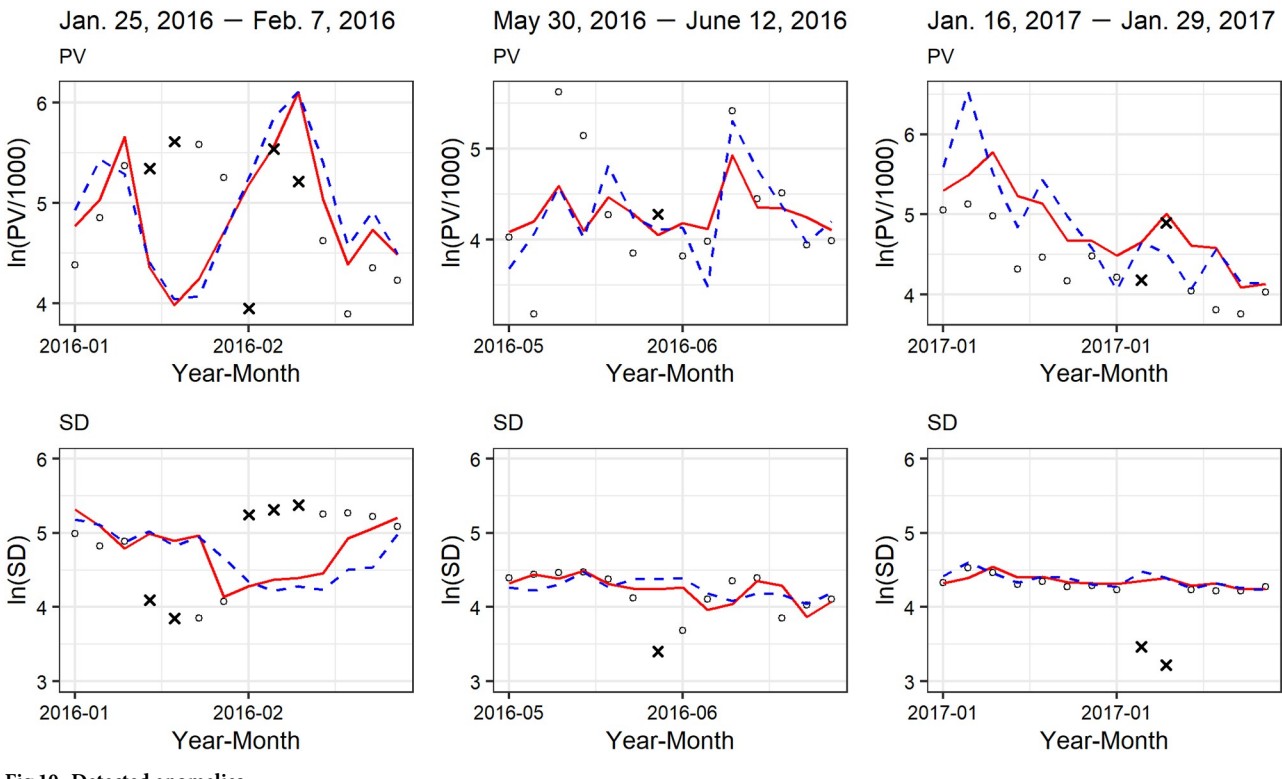

**Fig 10. Detected anomalies.**

tendency is particularly strong in SD, it seems that our method mainly detects anomalies in SD rather than PV.

Unlike the case of the evaluation section, we have no anomaly labels in this application. Therefore, we cannot evaluate our method's performance here, i.e., whether or not the detected anomalies are actually anomalous. Because the detected anomalies are expected to be caused by factors external to the Web advertising agency, we attempt to discuss the relationship between the detected anomalies and external factors to the web advertising agency such as the marketing activities of other physical advertising agencies. Here, we consider two external factors, namely delivering e-mail newsletters and running TV commercial spots.

First, other advertising agency delivered e-mail newsletters basically on every Wednesday from February 6, 2016 to January 14, 2018. We show PV via e-mail newsletters by day of the week in Fig 11. For each day of the week, we show minimum, maximum, and average values. We can see that, on average, PV via e-mail newsletters increases on Wednesday. It indicates that delivering e-mail newsletters tend to increase PV on Wednesday. Given this observation, we can expect that Wednesdays are likely to be detected as anomalies caused by e-mail newsletter deliveries. However, as shown in Table 4, we find that Wednesdays are less likely to be detected as anomalies: January 25, 2017 is the only Wednesday detected within the period of e-mail newsletter deliveries. In Fig 12, we show the filtered values of day-of-week component for PV. As in Fig 11, we show minimum, maximum, and average values. It reveals a quite similar pattern to Fig 11. These figures indicate that the effect of scheduled e-mail newsletter deliveries on PV is not large enough to be detected as anomalies and is captured by a day-of-week component in the model, which explains why Wednesdays are less likely to be detected as anomalies.

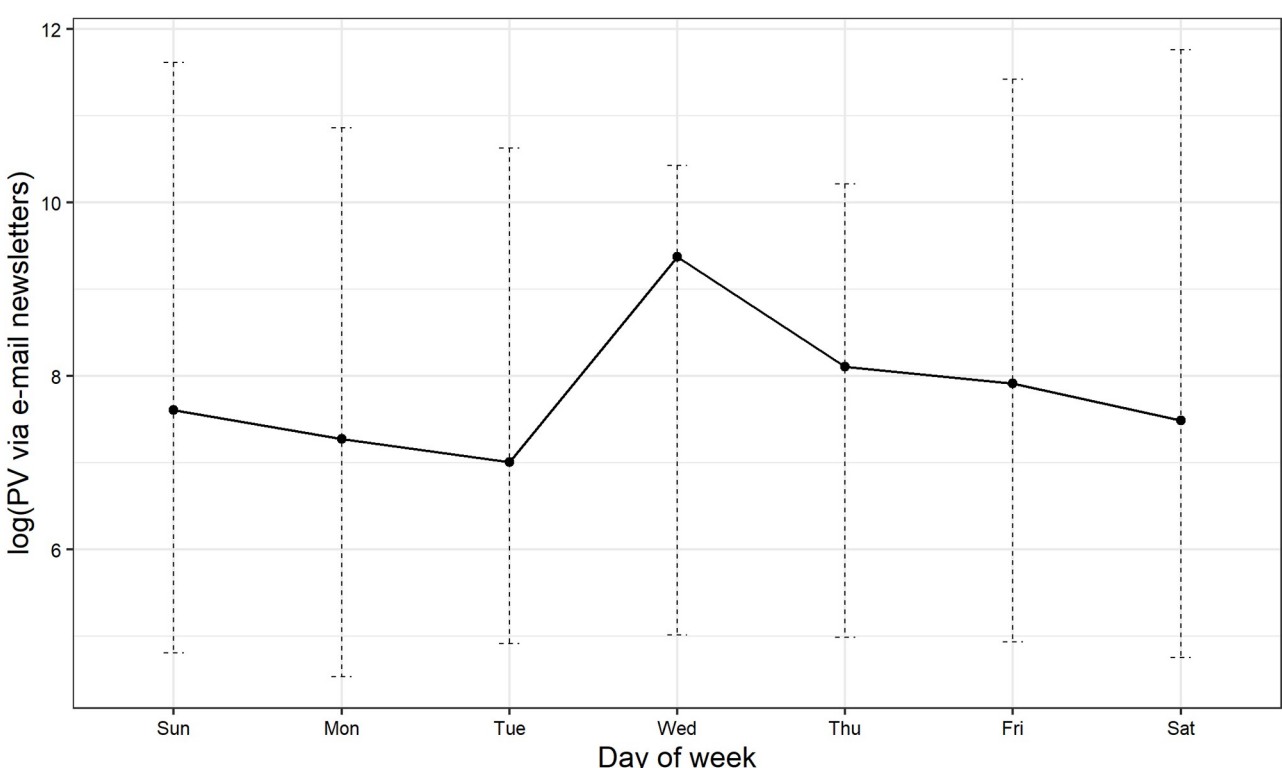

**Fig 11. PV via e-mail newsletter by day of week.**

Nevertheless, why are data on January 25, 2017 detected as anomalies? From the right two panels in Fig 10, we can observe that SD is far from the one-step ahead forecast of observations, while PV is close to it. It indicates that the increase in PV by e-mail newsletter deliveries is captured by the day-of-week component in the model. However, there is a significant decrease in SD beyond the scope of the forecast and our method detects the decrease in SD. This result suggests the importance of the simultaneous monitoring of more than one time series because we may miss this anomaly if we monitor PV only. We can explain the increase in PV and the decrease in SD by the increase in bounce rate. The bounce rate is defined as the percentage of visitors who enter the site and then leave rather than continue viewing other pages within the same site. Even if PV increases by e-mail newsletter deliveries, SD decreases if many visitors are single-page visitors. Such opposing movement of PV and SD suggests that the increase in

**Table 4. List of days detected as anomalies.**

| Day of week | Date |
|---|---|
| Thursday | January 28, 2016 |
| Friday | January 29, 2016 |
| Monday | February 1, 2016 |
| Tuesday | February 2, 2016 |
| Wednesday | February 3, 2016 |
| Sunday | June 5, 2016 |
| Tuesday | January 24, 2017 |
| Wednesday | January 25, 2017 |

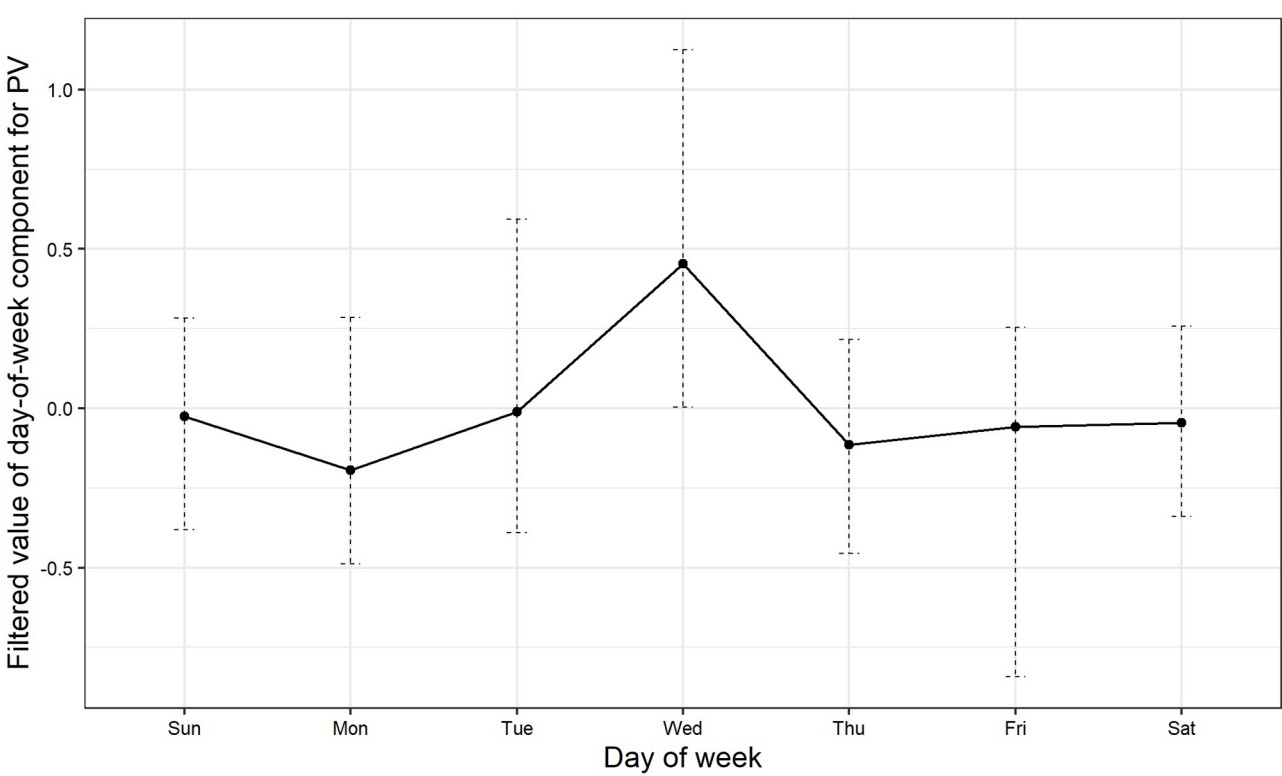

**Fig 12. Filtered value of day-of-week component.**

PV caused by e-mail newsletter deliveries is less likely to contribute to completing an insurance contract. In addition to January 25, 2017, there are two detected anomalies within the period of e-mail newsletter deliveries (June 5, 2016 and January 24, 2017). They might also be caused by e-mail newsletter deliveries because (1) the e-mail newsletters are also delivered on days of the week other than Wednesday, and (2) they reveal a similar pattern to that of January 25, 2017. The remaining detected anomalies cannot be explained by e-mail newsletter deliveries because they are outside the period of e-mail newsletter deliveries.

Second, the TV commercial spots produced by the other advertising agency were broadcast four times within the period of our test data, which is highlighted by a gray zone in Fig 9. We find that no anomalies are detected within the periods in which TV commercial spots were run. It seems that running TV commercial spots hardly contributed to PV and SD because the EC site in this study mainly deals with life-insurance goods and its TV commercial spots often focus on its product image.

We show the number of detected anomalies by threshold $\underline{r}$ in Fig 13. We can see that the number of detected anomalies decreases as the threshold $\underline{r}$ increases and the result is relatively robust. Although the number of detected anomalies is 107, about 15% of the observations, and looks quite large when $\underline{r} = 1$, it is not a very large value because we assume normal distribution: about 68% of the observations drawn from a normal distribution $\mathcal{N}(\mu, \sigma^2)$ are within $\mu \pm \sigma$ and about 95% of the observations are within $\mu \pm 2\sigma$. Based on this comparative statics, we choose $\underline{r} = 3.0$ to show the application result described above. Actually, it is not clear whether $\underline{r} = 3.0$ is the best value for this application. Finding an optimal threshold is not a straightforward task in general. Therefore, at present, an expert should set the threshold value depending on the data to be applied. For example, one can set a large threshold for noisy time series data,

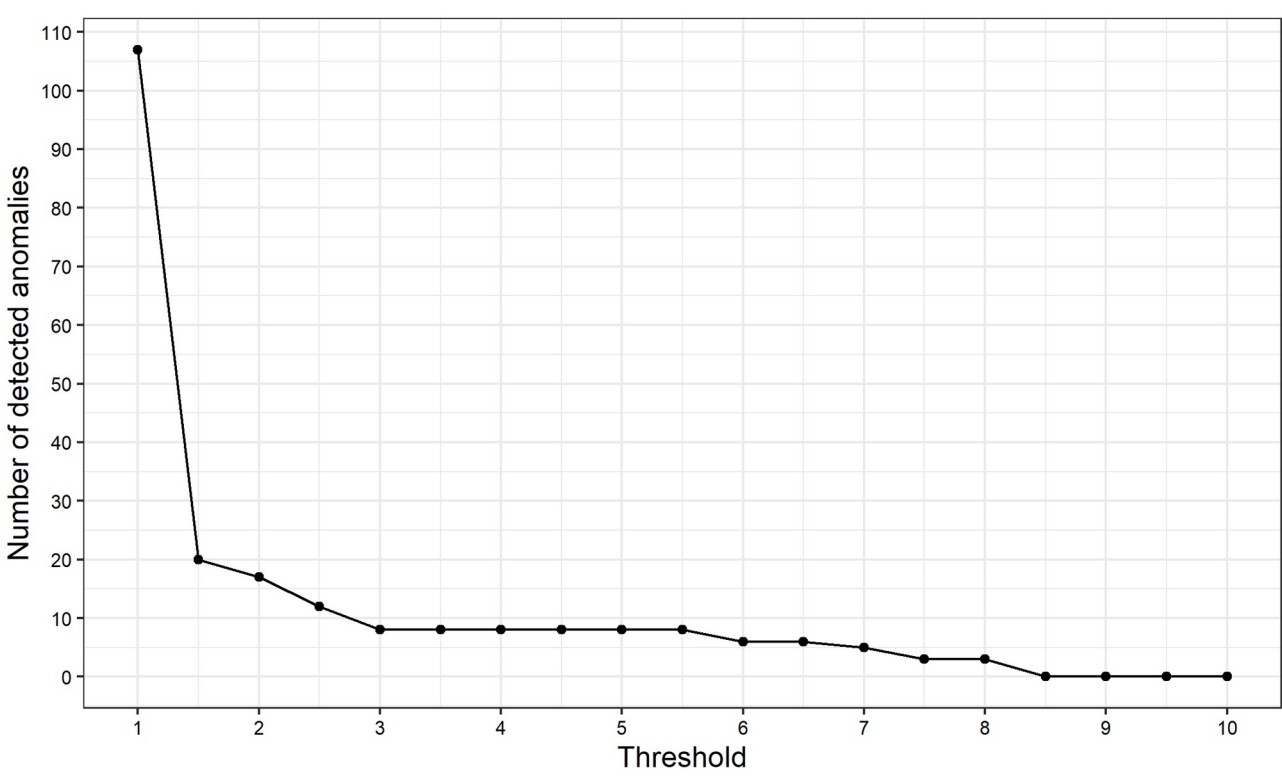

**Fig 13. Number of detected anomalies by threshold _r_.**

and vice versa. Alternatively, it might be possible to treat the threshold as a function of state variables. A systematic determination of the threshold is our future task.

## Conclusion and future tasks

We proposed a simple anomaly detection method that is applicable to unlabeled and non-stationary time series data and is sufficiently tractable, even for non-technical entities, by employing the density ratio estimation and the DLM. We evaluated our method using the publicly available benchmark data sets, the Yahoo A1 benchmark and the NAB realAdExchange data set. We find that our method achieves better or comparable performance compared to the existing methods which have been evaluated by using the same benchmark data sets. This is mainly because of the potential of our method to take into account larger number of explanatory variables in the model than the existing methods. The evaluation results imply that it is essential in time series anomaly detection to incorporate the specific information on time series data into the model. Moreover, we show our method's applicability to unlabeled real-world data by applying a bivariate version of the method to daily PV and SD data on an EC site. We can detect the observations suspected as anomalies and partially find out the cause of detected anomalies by utilizing the information external to the Web advertising agency. By discussing the relationship between the detected anomalies and e-mail newsletter deliveries, we find that the increase in PV caused by e-mail newsletter deliveries is less likely to contribute to completing an insurance contract. The result also suggests the importance of the simultaneous monitoring of more than one time series. However, we could not find the cause of anomalies detected outside the period of delivering e-mail newsletters and running TV commercial spots. Although it is suggested that other external factors exist, we cannot uncover the cause of

these anomalies based on the information at hand alone. As in our application, the cause of anomalies for less-informative entities in actual application is probably not clear in advance. Our method helps such entities to ask relevant business entities for the cause of anomalies by sequentially performing anomaly detection and detecting anomalies in a timely fashion.

In our method, we have two exogenous parameters to preliminarily set: the parameters in the NULL model and the threshold $r$ in the anomaly detection rule. As described in the application section, these two parameters play the same role in anomaly detection. Therefore, in actual application, it would be preferable to focus on the choice of a value of the threshold $r$ because all we need in terms of the parameters in the NULL model is to set their values such that the dispersion of the NULL model is greater than that of OUR model. Future tasks include constructing a systematic way to set the value of the threshold $r$, while we need to determine its value by trial and error at present.

## Supporting information

**S1 Appendix. State space representation.**
(PDF)

**S1 Table. Evaluation results on all time series in the Yahoo A1 benchmark.** We show the optimal threshold, $F_1$ score, precision, and recall on 59 time series in the A1 benchmark under each case. If the $F_1$ score is undefined under any $k$, we enter NA into the optimal threshold, $F_1$ score, precision, and recall.
(PDF)

**S2 Table. Results of applying local level model to the Yahoo A1 benchmark.** We show the optimal threshold, $F_1$ score, precision, and recall on 59 time series in the A1 benchmark under each case when we apply the local level model. If the $F_1$ score is undefined under any $k$, we enter NA into the optimal threshold, $F_1$ score, precision, and recall.
(PDF)

**S3 Table. Evaluation results on all time series in the NAB realAdExchange.** We show the optimal threshold, $F_1$ score, precision, and recall on 5 time series in the realAdExchange data set under each case. If the $F_1$ score is undefined under any $k$, we enter NA into the optimal threshold, $F_1$ score, precision, and recall.
(PDF)

## Acknowledgments

We are grateful to Flat inc. for sharing the data set with us. We also thank the participants at the International Conference on Operations Research 2019 for their helpful comments.

## Author Contributions

**Conceptualization:** Kei Takahashi.

**Data curation:** Kei Takahashi.

**Formal analysis:** Keisuke Yoshihara.

**Funding acquisition:** Kei Takahashi.

**Investigation:** Keisuke Yoshihara.

**Methodology:** Kei Takahashi.

**Project administration:** Kei Takahashi.

**Resources:** Kei Takahashi.

**Software:** Keisuke Yoshihara.

**Supervision:** Kei Takahashi.

**Validation:** Keisuke Yoshihara.

**Visualization:** Keisuke Yoshihara.

**Writing – original draft:** Keisuke Yoshihara.

**Writing – review & editing:** Keisuke Yoshihara.

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
