## [Decision Letter · Decision Letter 0]

19 Aug 2021

PONE-D-21-21491

A simple method for unsupervised anomaly detection: An application to Web time series data

PLOS ONE

Dear Dr. Yoshihara,

Thank you for submitting your manuscript to PLOS ONE. After careful consideration, we feel that it has merit but does not fully meet PLOS ONE’s publication criteria as it currently stands. Therefore, we invite you to submit a revised version of the manuscript that addresses the points raised during the review process.

We look forward to receiving your revised manuscript.

Kind regards,

Lianmeng Jiao

Academic Editor

PLOS ONE

Journal Requirements:

2. In your Methods section, please include additional information about your real dataset and ensure that you have included a statement specifying whether the collection method complied with the terms and conditions for the website. Please ensure that the links to both datasets are included in both the Data availability statement and the Methods section.

Reviewers' comments:

Reviewer's Responses to Questions

**Comments to the Author**

1. Is the manuscript technically sound, and do the data support the conclusions?

Reviewer #1: Yes

Reviewer #2: Partly

2. Has the statistical analysis been performed appropriately and rigorously? 

Reviewer #1: Yes

Reviewer #2: Yes

3. Have the authors made all data underlying the findings in their manuscript fully available?

Reviewer #1: Yes

Reviewer #2: Yes

4. Is the manuscript presented in an intelligible fashion and written in standard English?

Reviewer #1: Yes

Reviewer #2: Yes

5. Review Comments to the Author

Reviewer #1: This article proposed a simple anomaly detection method that is applicable to unlabeled time series data and is sufficiently tractable, even for non-technical entities, by using the density ratio estimation based on the state space model. The anomaly detection method is simple indeed.

Main concerns

1.The innovation of the article is not strong. It feels that it is an ordinary statistical method. It is abnormal if it is greater than R times the standard deviation?

2.In Table 1, Why is our method better than others? What are the advantages? What is the essential difference between these methods? Just because it's simple?

3.How to select R is not clear.

4.In application part, in other data sets, it is proved that our method can be used, but it is not compared with other methods, which can not explain the advantages of our method.

Reviewer #2: The paper presents a simple anomaly detection method for time-series data, by computing the ratio of the likelihood of the target model and dispersed model using a dynamic linear model. The events can be detected by comparing the value with a threshold. They applied the proposed method to Yahoo S5 dataset and Web time series data. The reviewer has several questions and concerns to be clarified.

The proposed method assumes a normal distribution to realize the log likelihood function and compute the ratios (and sequential learning by taking the difference between n and n-1 step). All the simplicity of the method comes from the assumption. However, such strong assumption should result in inaccurate estimation in the real world. Although the authors displayed experimental results using real-world dataset, they needed to discuss more about the problems and failure cases and show more experimental results with various datasets. Furthermore, please explain why the simple method can outperform other complex schemes. Further, it seems to be ad-hoc to choose the threshold (r) during detection. What are the values used in the experiments and how the values can be determined, systematically?

6. PLOS authors have the option to publish the peer review history of their article (what does this mean?). If published, this will include your full peer review and any attached files.

Reviewer #1: No

Reviewer #2: No

---

## [Author Response · Author response to Decision Letter 0]

4 Oct 2021

We provide our response to the editor and the reviewers as the separate file labeled "Response to Reviewers".

---

## [Decision Letter · Decision Letter 1]

27 Oct 2021

PONE-D-21-21491R1A simple method for unsupervised anomaly detection: An application to Web time series dataPLOS ONE

Dear Dr. Yoshihara,

Thank you for submitting your manuscript to PLOS ONE. After careful consideration, we feel that it has merit but does not fully meet PLOS ONE’s publication criteria as it currently stands. Therefore, we invite you to submit a revised version of the manuscript that addresses the points raised during the review process.

The reviewers still have some major concerns for your work. Please carefully revise this paper according to the comments.

We look forward to receiving your revised manuscript.

Kind regards,

Lianmeng Jiao

Academic Editor

PLOS ONE

Reviewers' comments:

Reviewer's Responses to Questions

**Comments to the Author**

1. If the authors have adequately addressed your comments raised in a previous round of review and you feel that this manuscript is now acceptable for publication, you may indicate that here to bypass the “Comments to the Author” section, enter your conflict of interest statement in the “Confidential to Editor” section, and submit your "Accept" recommendation.

Reviewer #3: (No Response)

2. Is the manuscript technically sound, and do the data support the conclusions?

Reviewer #3: No

3. Has the statistical analysis been performed appropriately and rigorously? 

Reviewer #3: Yes

4. Have the authors made all data underlying the findings in their manuscript fully available?

Reviewer #3: Yes

5. Is the manuscript presented in an intelligible fashion and written in standard English?

Reviewer #3: Yes

6. Review Comments to the Author

Reviewer #3: 1. This article lack of novelty. What are the advantages and major contributions of this method? The likelihood is usually used in detection. What is the difference between other methods?

2. The explanation why the simple method outperforms other complex schemes is not confidence. It is better to give more believable and confident explanations based on data and experience.

3. The performance may be largely dependent on the distribution and R-value. The normal distribution assumption lacks sufficient evidence from data and experience. Would the method based on other distributions perform better?

7. PLOS authors have the option to publish the peer review history of their article (what does this mean?). If published, this will include your full peer review and any attached files.

Reviewer #3: No

---

## [Author Response · Author response to Decision Letter 1]

10 Dec 2021

We provide our comments and revisions as the separate file labeled "Response to Reviewers".

---

## [Editor Report · Decision Letter 2]

26 Dec 2021

A simple method for unsupervised anomaly detection: An application to Web time series data

PONE-D-21-21491R2

Dear Dr. Yoshihara,

We’re pleased to inform you that your manuscript has been judged scientifically suitable for publication and will be formally accepted for publication once it meets all outstanding technical requirements.

Kind regards,

Lianmeng Jiao

Academic Editor

PLOS ONE

---

## [Editor Report · Acceptance letter]

3 Jan 2022

PONE-D-21-21491R2 

A simple method for unsupervised anomaly detection: An application to Web time series data 

Dear Dr. Yoshihara:

I'm pleased to inform you that your manuscript has been deemed suitable for publication in PLOS ONE. Congratulations! Your manuscript is now with our production department. 

Kind regards, 

on behalf of

Dr. Lianmeng Jiao 

Academic Editor

PLOS ONE